# Structural and Functional Support by Left Atrial Appendage Transplant to the Left Ventricle after a Myocardial Infarction

**DOI:** 10.3390/ijms23094661

**Published:** 2022-04-22

**Authors:** Jussi V. Leinonen, Päivi Leinikka, Miikka Tarkia, Milla Lampinen, Avishag K. Emanuelov, Ronen Beeri, Esko Kankuri, Eero Mervaala

**Affiliations:** 1Department of Pharmacology, Faculty of Medicine, University of Helsinki, Haartmaninkatu 8, P.O. Box 63, 00014 Helsinki, Finland; jussi.leinonen@mac.com (J.V.L.); paivi.leinikka@helsinki.fi (P.L.); miikka.tarkia@gmail.com (M.T.); milla.lampinen@helsinki.fi (M.L.); esko.kankuri@helsinki.fi (E.K.); 2Cardiovascular Research Centre, Heart Institute, Hadassah-Hebrew University Medical Centre, Jerusalem 91120, Israel; aemanuelov@hadassah.org.il (A.K.E.); rbeeri@hadassah.org.il (R.B.)

**Keywords:** adult stem cells, macrophages, atrial appendage, heart failure, tissue transplant

## Abstract

The left atrial appendage (LAA) of the adult heart has been shown to contain cardiac and myeloid progenitor cells. The resident myeloid progenitor population expresses an array of pro-regenerative paracrine factors. Cardiac constructs have been shown to inhibit deleterious remodeling of the heart using physical support. Due to these aspects, LAA holds promise as a regenerative transplant. LAAs from adult mT/mG mice were transplanted to the recipient 129X1-SvJ mice simultaneously as myocardial infarction (MI) was performed. A decellularized LAA patch was implanted in the control group. Two weeks after MI, the LAA patch had integrated to the ventricular wall, and migrated cells were seen in the MI area. The cells had two main phenotypes: small F4/80+ cells and large troponin C+ cells. After follow-up at 8 weeks, the LAA patch remained viable, and the functional status of the heart improved. Cardiac echo demonstrated that, after 6 weeks, the mice in the LAA-patch-treated group showed an increasing and statistically significant improvement in cardiac performance when compared to the MI and MI + decellularized patch controls. Physical patch-support (LAA and decellularized LAA patch) had an equal effect on the inhibition of deleterious remodeling, but only the LAA patch inhibited the hypertrophic response. Our study demonstrates that the LAA transplantation has the potential for use as a treatment for myocardial infarction. This method can putatively combine cell therapy (regenerative effect) and physical support (inhibition of deleterious remodeling).

## 1. Introduction

Myocardial infarction (MI) is a common cause of heart failure (HF), which is one of the leading causes of death in Western countries [1]. After an MI, a permanent fibrotic scar is formed, leading to deleterious ventricular remodeling and the contractile demise of the heart. As heart transplantation is currently the only viable treatment for heart failure, novel therapies are being pushed forward with considerable effort. The mammalian heart can regenerate cardiomyocytes during the neonatal period, but loses this ability soon after birth [2,3]. Two main strategies are used to achieve regeneration and diminish the fibrotic response in the failing heart: cardiac gene therapy and cardiac stem cell/tissue transplantation [4]. The results of meta-analyses have demonstrated that cell therapy has a consistent therapeutic effect on cardiac function after MI and encourages large-scale clinical studies [5]. The mechanisms behind the observed improvement in heart function after cell therapy administration have been under debate. Recent long-term studies demonstrate that a number of cells remain in the tissue, but they do not differentiate to cardiomyocytes, a strong suggestion for a mechanism via, for example, paracrine actions [6,7,8,9].

Following cardiac tissue injury, such as MI, the immune system plays an important role in acute inflammatory response and the following regenerative or fibrotic response [10]. Macrophages are an integral part of the regenerative response in mammals. Immune system development and regenerative response are closely intertwined [11]. It has been shown that mouse neonates that were depleted of macrophages were unable to regenerate myocardium [12]. Specifically, the resident, embryonic-derived macrophage subsets have been shown to have pro-regenerative capabilities that were lost during maturation or bone marrow activation after tissue injury [13,14]. Macrophages are also shown to be crucial for myocardial healing through a fibrotic response in the adult heart [15]. In addition, the essential functions of resident macrophages have been revealed for normal heart homeostasis. During cardiac development, macrophages are required for coronary vessel development and function [16], and it has been shown that tissue-resident macrophages are important mediators of electrical conduction in the adult heart [17].

The atrial appendages are often considered as mere volume reservoirs of the heart and part of the atria. However, they have several unique features, which differentiate them from the atria [18,19]. In clinical medicine, the left atrial appendage (LAA) presents with some problematic features, because it is the most common site for thrombus formation in the setting of atrial fibrillation, and because the structure of the LAA is complex, making ultrasound evaluation difficult [20]. Our studies on the tissue composition of the murine LAA at the cellular level revealed cellular potential for cardiac regeneration in the LAA, putatively stemming from de novo cardiomyocyte formation and/or the paracrine effect by resident heart tissue macrophages [21,22]. Atrial-appendage-derived cells have been studied, by us and others, in animal models [23] and humans with promising results [24,25,26,27]. Atrial appendage tissue or micrograft transplants provide an autologous cardiac cell source and include all cardiac cell populations with their relevant extracellular matrix support. Moreover, they have a high capacity to produce paracrine therapeutic factors [22,26]. They can also be readily clinically harvested during open heart surgery.

The detached stem cells that are injected into the hostile microenvironment after MI have a very low survival rate. Engineered heart tissue skills have overcome many of the problems related to the cell injections [28,29]. The results improved when multicellular preparations were used, and vascularization increased in the grafts [30]. An interesting option is to seed the cells to a decellularized native myocardial matrix, which enables the bioengineering of functional human myocardial-like tissue of multiple complexities [31]. A much less-used approach is to perform whole-tissue transplantation, either autologously or non-autologously. In this study, we demonstrate that the whole LAA transplantation inhibits scar formation and adverse remodeling of the heart after an MI. The transplant-derived cells migrate to the host tissue and remain viable in the tissue at least for 8 weeks.

## 2. Results

Based on our previous findings, we hypothesized that the LAA could serve as a regenerative tissue transplant after an MI [21,22,23,24,25,26,27]. The LAA is thin, exhibits a complex structure, and boasts many trabeculae, thus resembling ventricles more than the atria (Figure 1A,B and Appendix A). In this study, we performed a transplantation of the LAA from mTomato+ mouse to the nearest background strain (129X1-SvJ) mouse to track the cell survival and migration without using immunosuppressants (Figure 1C and Appendix A). Simultaneously, an MI was performed. The operation is described in more detail in the Appendix A (Appendix A). We have also successfully performed an autologous transplantation of the LAA in a murine model by ligating the LAA through an upper thoracotomy.

Within one or two weeks after the operation, the transplanted LAA was viable and attached to the ventricular wall (Figure 1D,E). Three-dimensional imaging using the iDisco protocol revealed a border zone on the mTomato+ transplant and a considerable infarct scar. Some cells were seen migrating to the infarct area (Figure 1F–H and Appendix A). A closer post-operational histological analysis revealed clusters of migrating cells with different phenotypes as early as 1 week after the operation (Figure 2A–D). After 1 week, immunostaining revealed two main cellular phenotypes in migrating transplant-originated cells: troponin C positive myocyte-like cells and F4/80 positive macrophages. At 2 weeks, the LAA explant is solidly integrated to the left ventricle. Most of the migrated mTomato+ cells express F4/80, a macrophage marker (Figure 2E–H).

After promising initial short-term observations, we performed a long-term follow-up at 8 weeks. Three different controls were used. The main control group received a decellularized LAA transplant after MI. The other two groups were MI and sham. Both LAA patches and decellularized LAA patches had a comparable size (Figure 3A). In the sham group, the pericardium was cut open. We performed an echocardiography follow-up to investigate functional differences between the groups and to verify the initial similarity between the treatment (*n* = 6) and main control group (*n* = 6). Two weeks after the operation, the treatment group (LAA patch) and the main control group (decellularized patch) demonstrated similar characteristics using several functional parameters (Figure 3B–G). This confirmed a good standardized operational outcome, with comparable MI sizes. At 4 weeks, there was no statistically significant improvement in the functional status of the heart in the treatment group, when compared to the decellularized patch control group. From 4 to 8 weeks, the contractility of the anterior wall of the left ventricle (patch-attached area) in the LAA patch transplant group demonstrated a gradual improvement, which was obvious, even when using an approximate on-site echo examination (Appendix A). This improvement was only seen in the treatment group. At 7 weeks, there was a statistically significant difference seen in longitudinal strain (long-axis) and circumferential strain (short-axis) measurements (Figure 3C,D). At 8 weeks, the improvement of the ejection fraction (EF) reached statistical significance as well (Figure 3B). Interestingly, the structural support gained from the decellularized patch similarly contributes to the inhibition of adverse ventricular remodeling, as demonstrated by diastolic (EDV) and systolic (ESV) ventricular volume measurements (Figure 3F,G).

After 8 weeks, we sacrificed the mice and analyzed the whole-mount hearts using iDisco 3D imaging. We performed a double staining using RFP (mTomato) and troponin C antibodies. We evaluated the amount of scar tissue (MI size) by subtracting the troponin-C-positive tissue volume from total left ventricular tissue volume (Figure 4A). The amount of scar tissue was significantly lower in the treatment group (*n* = 6) when compared to the decellularized patch control group (*n* = 3). We also calculated the total heart tissue volume to compare the amount of hypertrophic response in different groups (Figure 4B). The hypertrophic response was significantly attenuated in the treatment group. The number of migrated mTomato+ transplant-originated cells was similar between subjects in the treatment group; a small amount of unspecific signal was seen in the decellularized patch control group (Figure 4C). The mTomato+ LAA patch was visualized 8 weeks after transplantation and the migrated cells were concentrated in the infarct area/border zone (Figure 4D–G and Appendix A). A closer histological analysis revealed that the mTomato+ transplant is engulfed by the left ventricular wall 8 weeks after MI. The surviving transplant is mostly troponin-C-negative (Figure 5A–C). Many of the remotely located migrated cells express macrophage marker F4/80 (Figure 5D,E). Another significant portion of the mTomato+ cells were in vessel-like structures next to the infarct zone, together with some troponin C+ cells (Figure 5F,G).

## 3. Discussion

Atrial appendage transplantation has the potential to deliver autologous cardiac cells, including various progenitor cells, in their native environment to the injured tissue for cardiac regeneration [21,22,23,24,25,26,27]. In terms of the LAA, transplantation is further supported based on its characteristics; the most clinically relevant are its size and its potential for extensibility over the infarct area. In this preclinical study, we have provided evidence that the detached LAA has the potential to serve as a pro-regenerative source of autologous tissue in an acute ischemic setting.

We demonstrated that the LAA transplants can remain viable in the long-term, even without pre-vascularization procedures, and that the migrated cells from the transplant remain in the host tissue. The functional improvement in the hearts of the treatment group gained significant momentum 4 weeks after the transplantation operations. This could be related to the time required for neovascularization to reach its required maturity to enhance the blood flow in the ischemic heart. Only small areas of the transplanted mTomato+ LAA patch and some migrated cells remained troponin-C-positive 8 weeks after the operations. This suggests that factors other than the direct differentiation of transplanted cells are the main contributors to the functional improvement. The F4/80+ resident macrophages of the LAA are the source of a wide array of paracrine factors, including VEGF-A and IGF-1 [22]. Interestingly, the same observation of a secretome-mediated effect and an array of paracrine factors was seen in human atrial-appendage-derived cardiac progenitor cells [7,8,9,26]. The conclusion of a paracrine-mediated beneficial effect is further supported in the current study by the infarct area localization and the long-term survivability of F4/80+ transplant-derived migrated cells.

Previously, in a single study, a piece of the LAA was used as a tissue patch on top of the MI area [32]. The patch transplantation was performed 3 weeks after MI, and omentopexy [33] was performed to enhance transplant vascularization. At the 4-week follow-up, positive results were acquired on the contractile function of the heart and the inhibition of cardiac remodeling. The observed beneficial effect was considered to be caused mainly by paracrine activity, which is in accordance with our study. The atrial appendages are also a rich source of natriuretic peptides, for example, type A atrial natriuretic peptide (ANP), which have been demonstrated to exert antifibrotic and cardioprotective effects on the myocardium [34,35].

A major translational shortcoming of this study is that we performed LAA transplantation simultaneously with the MI, which is not applicable to a clinical setting. We decided to complete the trial using single thoracotomy, because in a mouse model, two thoracotomies in a week would have caused too high a mortality rate. However, our results are promising because we did not use an allogenic transplantation model or an immunosuppressant. This suggests that the functional outcome could be improved by using an autologous transplantation. Further investigation is required, preferably using a large animal model and, to more closely resemble a clinical setting, a closed-chest coronary balloon occlusion of the left anterior descending artery, followed by autologous LAA tissue or LAA micrograft transplantation a few days later.

## 4. Materials and Methods

### 4.1. Mouse Strains and Patch Transplantation

LAAs from adult mT/mG mice (JAX 007576, The Jackson Laboratory, Bar Harbor, ME, USA) were transplanted to the recipient 129X1-SvJ mice (JAX 000691) as myocardial infarction (MI) was performed. A decellularized LAA patch was implanted into the control group. Prolene (Ethicon Inc., Raritan, NJ, USA) 8-0 was used for LAD ligation and Vicyl Rapide (Ethicon) 7-0 for patch attachment. A detailed description of the operation is included in Appendix A.

### 4.2. Ethics

All animal experiments were conducted according to the European Community guidelines for the use of experimental animals. The experiments were approved by the Finnish National Animal Experiment Board (permit numbers ESAVI/6718/04.10.03/2012 and ESAVI/8054/04.10.07/2016).

### 4.3. Left Atrial Appendage Decellularization

To produce an acellular, extracellular matrix LAA patch for the control group treatment, ten LAAs were decellularised in 20 mL of 1% (*w*/*v*) sodium deoxycholate (Sigma-Aldrich, St. Louis, MO, USA, 264101) in PBS, pH 7.4 for 2 × 24 h under constant agitation at RT. Distinctly translucent LAAs were washed four times with 20 mL PBS during a 24-h incubation at RT to rinse out the detergent. Tissues were sterilised in cold, freshly prepared 0.1% (*w*/*w*) peracetic acid (Sigma-Aldrich, 433241) in PBS at pH 7.0 for 10 min, and stored in sterile PBS, pH 7.4, until transplantation.

### 4.4. Cardiac Ultrasound and Analysis

Serial cardiac ultrasound experiments were conducted under isoflurane anesthesia using Vevo 2100 (FUJIFILM VisualSonics Inc., Toronto, ON, Canada) by an experienced lab technician. Acquired data were analyzed using Vevo LAB Vevostrain 1.7.1 (FUJIFILM VisualSonics) software from the longitudinal and short axis perspective [36,37].

### 4.5. 3D Imaging (iDisco)

Before removing the heart, the mouse was first infused with PBS, followed by PFA-infusion. After that, iDisco staining and clearing were preformed, according to methods previously published [38]. The finalized whole-heart specimen (week 8) was imaged with a LaVision Ultramicroscope II lightsheet microscope (LaVision BioTec GmbH, Bielefeld, Germany). Samples from 1 and 2 weeks after operation were imaged using a Bioptonics 3001 OPT (Bioptonics Microscopy, Edinburgh, UK) scanner and optical projection tomography (OPT); tomographic data were reconstructed using N-Recon software (SkyScan, Kontich, Belgium). The acquired images were analyzed using Imaris 8 (Bitplane, Oxford Instruments plc, Abingdon, UK). The volume of infarcted tissue (scar) was calculated using surface rendering of the autofluorescence signal of the left ventricle subtracted with a troponin-C-positive section of the left ventricle. Migrated mTomato+ cells were calculated using Imaris 8 Spots object detection model. The threshold was standardized in all measurements.

### 4.6. Immunohistochemistry

Immunohistochemistry was performed on tissue sections cut from snap-frozen hearts. Air-dried 10-μm tissue sections were fixed for 3 min in ETOH 70%, 3 min in ETOH 95%, and 3 min in ETOH 100%. Primary antibodies were incubated overnight in +4 °C and secondary antibodies for 40 min in room temperature. Washes were performed using PBS. Prolong Gold antifade with DAPI (Thermo Fisher Scientific Inc., Waltham, MA, USA) was used as an imaging mountant.

### 4.7. Antibodies

Primary antibodies: anti-RFP (600-401-379 Rockland Immunochemicals Inc., Pottstown, PA, USA) 1:500, anti-Cardiac Troponin C (ab30807 Abcam, Cambridge, UK) 1:500, anti-F4/80 (ab6640 Abcam) 1:200. Secondary antibodies: Alexa Fluor 488, 594 and 647 (donkey-anti goat, rat and rabbit, Thermo Fisher Scientific, Waltham, MA, USA) were used at a concentration of 1:200. The same concentrations were used in the iDisco protocol and standard immunohistochemistry.

### 4.8. Statistical Methods

Statistical analysis between two groups was performed using two-tailed unpaired *t*-test; *p*-values < 0.05 were considered significant. Data are presented as mean ± SD.

## 5. Conclusions

In conclusion, our results provide evidence of effectiveness and insight into the molecular mechanisms of left atrial appendage transplantation for heart failure. The therapy can easily be clinically administered, either as an epicardial tissue transplant or as epicardial left atrial appendage micrograft transplants, when mechanically minced during open heart surgery, as reported previously by us using the right atrial appendage [24,25]. Given the excellent clinical applicability of atrial appendage epicardial therapy, further clinical trials evaluating therapy efficacy are warranted.

## Figures and Tables

**Figure 1 ijms-23-04661-f001:**
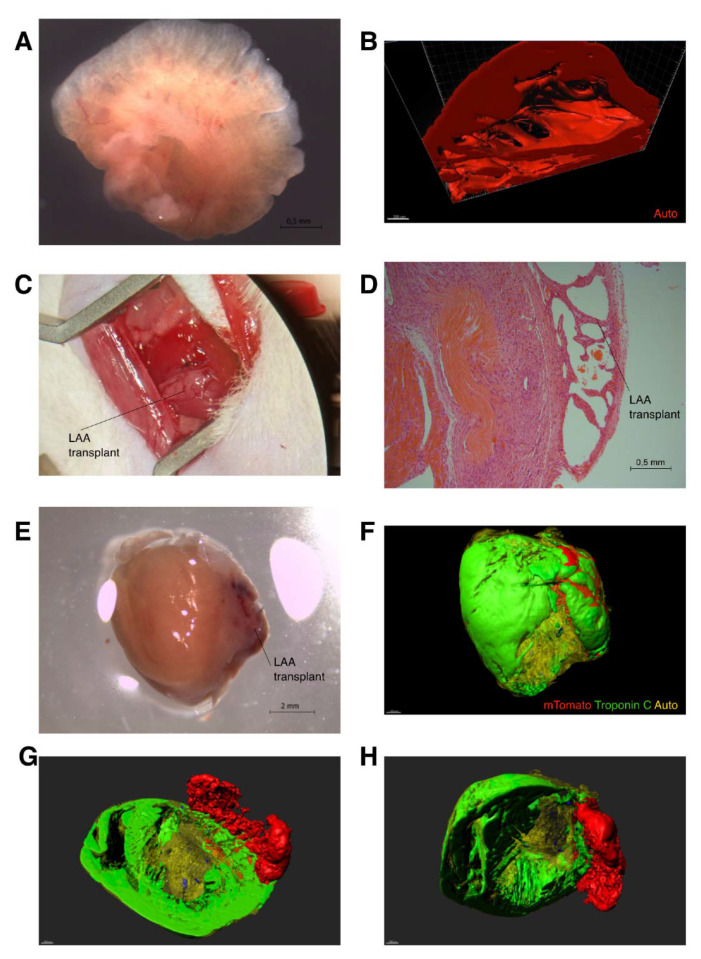
Left atrial appendage transplantation operation and 3D visualization of the transplant. (**A**) Visualization of a detached LAA before transplantation. (**B**) 3D image of the attached LAA using autofluorescence. (**C**) Visualization of the LAA attached to the border-zone of myocardial infarction. (**D**) H&E staining 2 weeks after operation showing LAA attached to the ventricular wall; the fibrotic scar is also seen. (**E**) LAA transplant (mTomato) with MI 2 weeks after operation. (**F**–**H**) Whole mount images of the same specimen using the iDisco clearing and immunostaining protocol. (mTomato = red, mTomato (migrated cells) = blue, troponin C = green, and autofluorescence = yellow).

**Figure 2 ijms-23-04661-f002:**
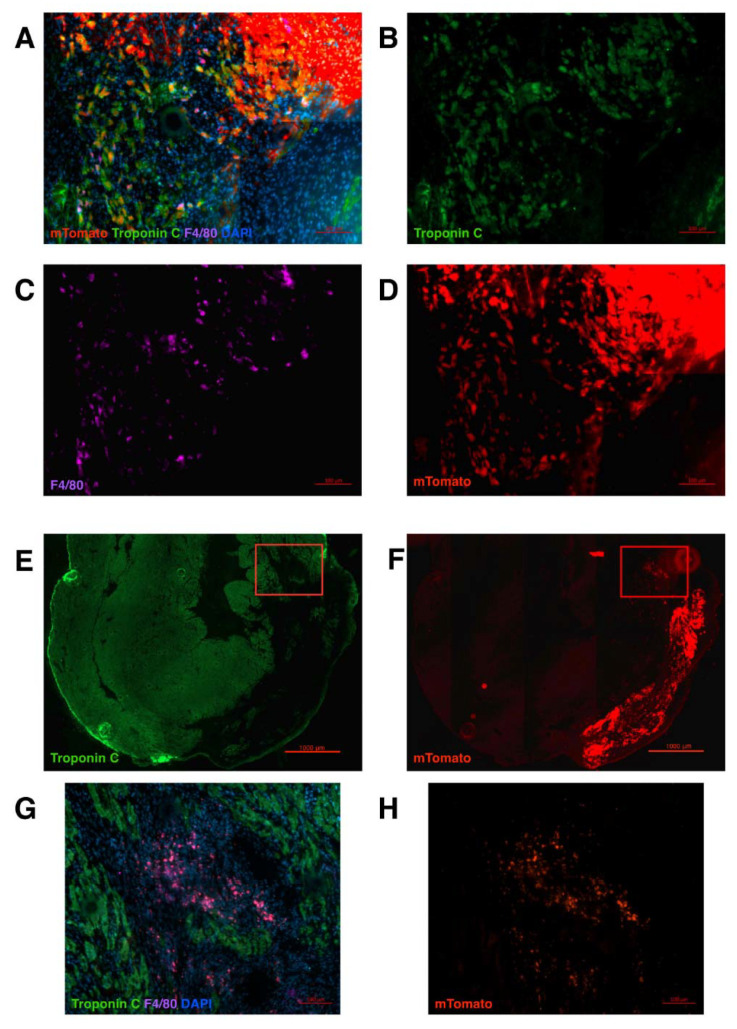
Tissue section analysis 1–2 weeks after the LAA transplantation. (**A**–**D**) 1 week after transplantation, a large number of cells is observed migrating from the transplant (mTomato = red). Most of the cells are stained with troponin C (green) or with the macrophage marker (F4/80 = magenta). (**E**,**F**) 2 weeks after transplantation, the LAA transplant (mTomato = red) is solidly integrated to the left ventricle. Some of the transplant-originated cells express troponin C (green). (**G**,**H**) Clusters of small transplant-originated cells are seen in the infarct zone further away from the transplant (magnification from the red frame in **E** and **F**). Migrated cells co-express F4/80 (magenta) with mTomato (red). DAPI = blue.

**Figure 3 ijms-23-04661-f003:**
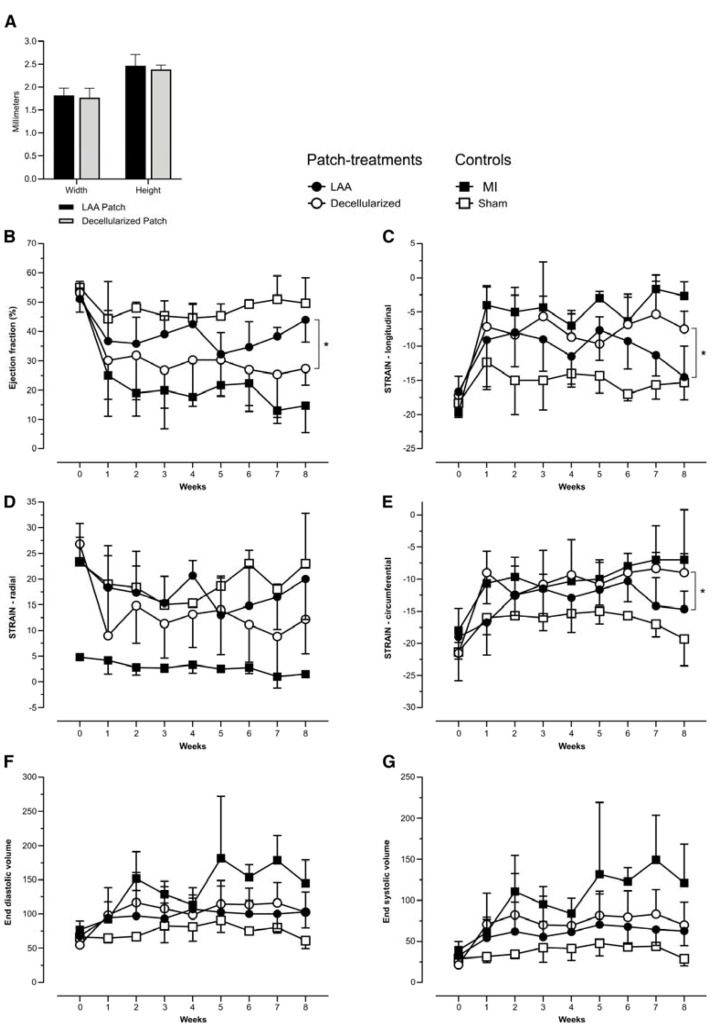
Patch comparisons and functional (echocardiography) follow-up after the LAA transplantation. (**A**) Comparison of the widths and heights of LAA and decellularized patches. (**B**) Ejection fractions were similar at 2 weeks, but during the follow-up, the difference between the two treatment groups increased, reaching a statistical significance at the 8w time point (*p* = 0.02). (**C**) The longitudinal strain analysis (long axis) demonstrates similar characteristics of two treatment groups before and 2 weeks after MI, but their difference reaches a statistical significance at 7 weeks (*p* = 0.03) and further increases at 8 weeks (*p* = 0.01). (**D**) The radial strain analysis (longitudinal axis) demonstrates an increasing difference between the two treatment groups, but statistical significance was not reached. (**E**) The circumferential strain analysis (short axis) demonstrates results similar to longitudinal strain. Statistical significance is at 7 weeks (*p* = 0.02) and at 8 weeks (*p* = 0.01). (**F**,**G**) End-diastolic and end-systolic volumes at indicated times after MI. LAA patch (*n* = 6), decellularized LAA patch (*n* = 6), MI (*n* = 3), and Sham (*n* = 3). * *p* < 0.05.

**Figure 4 ijms-23-04661-f004:**
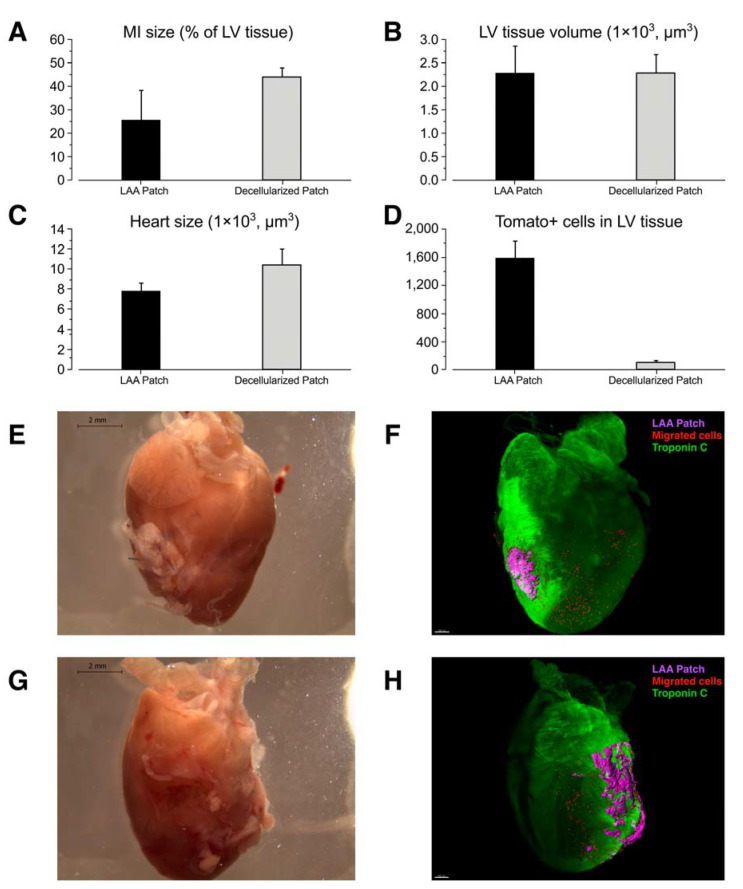
3D analysis 8 weeks after the LAA transplantation. (**A**) Scar tissue size comparison of the LAA-transplanted hearts (*n* = 6) vs. the decellularized patch controls (*n* = 6) demonstrating a smaller infarct area in the treated group. Scar volume is calculated by subtracting the troponin-C-positive signal from the autofluorescence of the left ventricle. Single measurements from the MI-group and Sham-group are shown as a reference. (**B**) Comparison of the size of the left ventricles after LAA transplants or the decellularized transplants before operation. (**C**) Heart tissue volume comparison of the LAA-transplanted-hearts vs. controls. Whole tissue volume of the hearts is calculated to evaluate the hypertrophic response. (**D**) Number of migrated mTomato+ cells compared to the control (false positive signal). (**E**–**H**) Whole mount images of the mTomato+ LAA-transplant-treated hearts. A comparison of light images and Imaris-processed 3D images. E + F and G + H are from the same specimen. Autofluorescence (green); cells migrated from the transplant (red); LAA patch (magenta).

**Figure 5 ijms-23-04661-f005:**
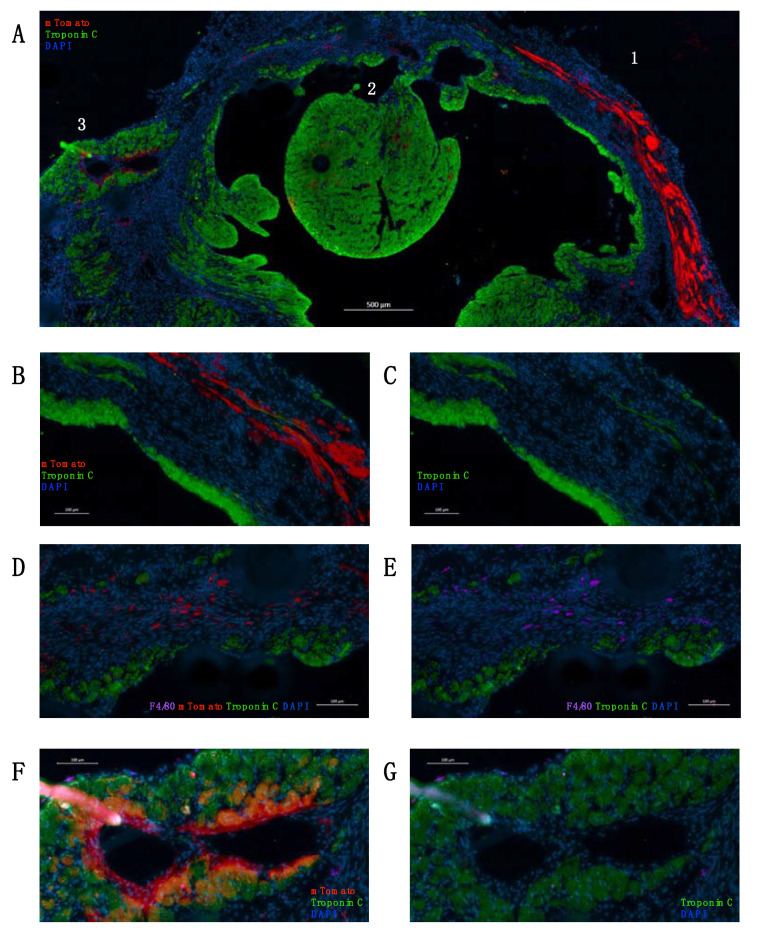
Tissue section analysis 8 weeks after the LAA transplantation. (**A**) Cross-sectional image from mitral valve level demonstrating a clear mTomato signal 8 weeks after operation. Troponin C = green, mTomato = red, and DAPI = blue. The following images are zoomed from frames 1, 2, and 3. (**B**,**C**) Frame 1: an image from LAA transplant zone demonstrating a strong mTomato signal, mostly in non-troponin-C-stained tissue. (**D**,**E**) Frame 2: an image from the remote infarct area demonstrating mTomato+ migrated cells, which co-express F4/80 (magenta). (**F**,**G**) Frame 3: an image from the remote infarct area demonstrating a strong mTomato signal in a vessel structure, together with some co-stained troponin-C-positive cells.

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
