# Peer review of "Structural and Functional Support by Left Atrial Appendage Transplant to the Left Ventricle after a Myocardial Infarction"

_ijms, 2022, doi:10.3390/ijms23094661_

Round 1
Reviewer 1 Report
In the paper "Left atrial appendage transplant integrates to the left ventricle after a myocardial infarction initiating structural and functional recovery" Leinonen et al. evaluated in an experimental mouse model of myocardial infarction the effect of left auricle administration on myocardial remodelling.
The study has a good research design and is well conducted, furthermore the paper is well written and the research results well represented
Minor point
- Please include some comments on the future prospects of this research
-A few sentences about the clinical applicability of this research could be useful to make the importance of the study clear.
Author Response
Thank you very much for your careful review and the constructive criticisms concerning our manuscript IJMS-167682. We have done our best to incorporate all the changes you recommended in this revision. Changes made to the manuscript have been marked up using the “Track Changes” function.
Reviewer’s minor points for the Author:
- Please include some comments on the future prospects of this research
- A few sentences about the clinical applicability of this research could be useful to make the importance of the study clear.
- English language and style are fine/minor spell check required
Authors:
As suggested by the Reviewer #1, we have now included a short paragraph at the end of the Discussion, where we 1) comment on the future prospects as well as 2) on the clinical applicability of our research.
- The manuscript has now undergone English language editing by MDPI. These changes have also been incorporated into the revised manuscript.
Reviewer 2 Report
The article is very well conceived and structured, the idea is innovative with a lot of practical implications and applications.
Starting from the hypothesis that the left atrial appendage of the adult heart has been shown to contain cardiac and myeloid progenitor cells which expresses a lot of pro-regenerative paracrine factors, the study demonstrates that the LAA transplantation has the potential to be used as a treatment for myocardial infarction. This complex method combine cell therapy (regenerative effect) and physical support (inhibition of deleterious remodeling after myocardial infarction). Because of those issues LAA could be an ideal tissue to be used as transplant. This experimental method could have an important clinical applicability.
The method concerning cardiac stem cell treatment in Myocardial Infarction is not very new, but the ideea of the article is original and interesting, reflecting the preoccupation of the authors. In this regard, on Research gate, In August 2013, European Heart Journal 34(suppl 1) (DOI: 10.1093/eurheartj/eht308.P1464) published a Conference entitled ”Cardiac progenitor cells from the left atrial appendage may originate from a resident non-hematopoietic myeloid progenitor population”, sustained by Jussi Leinonen, the first author of this manuscript.
I think the research is a serious one, well written, the method is very clear described.
The conclusions of the study are integrated in the discussion section, leaving the possibility of future research to contribute to the deepening of the researched topic.
The purpose of the research is well argued and justified.
Maybe the title could be revised, the title should not be a conclusion/sentence/statement
Author Response
Thank you very much for your careful review and the constructive criticisms concerning our manuscript IJMS-167682. We have done our best to incorporate all the changes you recommended in this revision. Changes made to the manuscript have been marked up using the “Track Changes” function.
Reviewer’s minor points for the Author:
- Maybe the title could be revised, the title should not be a conclusion/sentence/statement
- Moderate English changes required
Authors:
- The title has been revised. The revised title is “Structural and functional support by left atrial appendage transplant to the left ventricle after a myocardial infarction”.
- The manuscript has now undergone English language editing by MDPI. These changes have been incorporated into the revised manuscript.